# Alzheimer's disease image classification based on enhanced residual attention network

Xiaoli Li[1]*, Bairui Gong[2], Xinfang Chen[2], Hui Li[3], Guoming Yuan[1]

**1** School of Emergency Management, Institute of Disaster Prevention, Sanhe, Hebei, China, **2** School of Information Engineering, Institute of Disaster Prevention, Sanhe, Hebei, China, **3** School of Foreign Languages, Institute of Disaster Prevention, Sanhe, Hebei, China

* ms_lixiaoli@163.com

**Data Availability Statement:** All files are available from the URL:https://www.kaggle.com/code/bairuigong/alzheimers-dataset-eran-99.

**Funding:** This study was funded by the Langfang Bureau of Science and Technology, Hebei

## Abstract

With the increasing number of patients with Alzheimer's Disease (AD), the demand for early diagnosis and intervention is becoming increasingly urgent. The traditional detection methods for Alzheimer's disease mainly rely on clinical symptoms, biomarkers, and imaging examinations. However, these methods have limitations in the early detection of Alzheimer's disease, such as strong subjectivity in diagnostic criteria, high detection costs, and high misdiagnosis rates. To address these issues, this study proposes a deep learning model to detect Alzheimer's disease; it is called Enhanced Residual Attention Network (ERAN) that can classify medical images. By combining residual learning, attention mechanism, and soft thresholding, the feature representation ability and classification accuracy of the model have been improved. The accuracy of the model in detecting Alzheimer's disease has reached 99.36%, with a loss rate of only 0.0264. The experimental results indicate that the Enhanced Residual Attention Network has achieved excellent performance on the Alzheimer's disease test dataset, providing strong support for the early diagnosis and treatment of Alzheimer's disease.

## 1 Introduction

Alzheimer's Disease (AD) is a common neurodegenerative disease characterized by progressive cognitive decline, mainly affecting memory, thinking, and behavior. The traditional detection methods for Alzheimer's disease mainly rely on clinical symptoms, biomarkers, and imaging examinations [1]. However, these methods have limitations in the early detection of Alzheimer's disease, such as strong subjectivity in diagnostic criteria, high detection costs, and high misdiagnosis rates [2]. Sengul Dogan et al. proposed Lattice123 pattern, dynamic graph generation, and MDWT(Multilevel Discrete Wavelet Transform)-based multilevel feature extraction, which can detect AD accurately as the proposed pattern can extract subtle changes from the EEG(electroencephalogram) signal accurately and achieve significant results [3, 4]. In recent years, deep-learning-based Alzheimer's disease detection methods have received widespread attention and have been widely applied in the field of computer-aided analysis technology [5, 6]. Deep learning has achieved significant results in areas such as image classification,

Province, China in the form of a grant [2024011019] to XL.

**Competing interests:** The authors have declared that no competing interests exist.

object detection, and semantic segmentation [7, 8]. Particularly, Convolutional Neural Networks (CNN) have shown strong performance in medical image analysis [9–11]. In addition, Deep Residual Shrinkage Networks (DRSN) have shown superior performance in tasks such as fault diagnosis and image recognition [12]. Yu Song et al. combined Convolutional Neural Networks (CNN) technology with magnetic resonance imaging technology to design a 3D ResNet algorithm for AD classification, achieving an accuracy of 97.43% on the test set [13]. Zhu Jianbo et al. proposed a convolutional neural network FAMENET that integrated adaptive attention mechanism and data augmentation technology, which has alleviated data imbalance by introducing data augmentation technology, Focal Loss loss function, as well as adaptive attention mechanism to solve the problem of information loss caused by feature extraction and down-sampling in input images. Through extensive comparative experiments on public datasets, the classification accuracy of FAMENET only reached 79.95% [14]. Mingfeng Jiang et al. proposed a new method based on the external-attention mechanism for image classification of Alzheimer's disease, whose experimental results have shown that the new method can effectively improve the classification performance, and the accuracy of the fusion model MLP-C is 98.73% [15]. Wang Bin et al. designed a classification network for Alzheimer's disease (AD), Mild Cognitive Impairment (MCI), and Client of Normal (CN) based on an improved ResNet to address the difficulty in recognizing and classifying brain magnetic resonance imaging (MRI) in the three-stage population. On the randomly divided dataset, the classification accuracy of AD/MCI/CN reached 83.54%, the accuracy of AD/CN 95.27%, and the accuracy of MCI/CN 85.07% [16]. Ela Kaplan et al. presented a machine learning model called feed-forward Local Phase Quantization Network (LPQNet); LPQNet attained an accuracy of 99.62% on the Kaggle AD dataset using four classes [17].

Based on the above, existing algorithms for processing complex medical images, such as MRI scans for Alzheimer's disease, often fail to fully capture subtle pathological changes, leading to suboptimal accuracy. While certain methods achieve high accuracy, they heavily rely on manual feature extraction, which is time-intensive and restricts the model's generalizability and practical applicability [18]. Furthermore, challenges such as data imbalance and model interpretability remain significant concerns [19]. To address these issues, this study introduces a deep learning model, the Enhanced Residual Attention Network (ERAN), designed specifically for the classification of MRI images in Alzheimer's disease diagnosis. The model is supposed to be compared with other models such as Attention_Unet, Tiny-VIT, RSN, and MobileNet V3 on the same dataset. ERAN achieves an accuracy of 99.36% on the test set and a loss rate of only 0.0264. In terms of performance, it is significantly superior to other models and achieves a more accurate detection of Alzheimer's disease MRI images. Experiments have shown that Enhanced Residual Attention Network (ERAN) aims to improve the model's feature representation ability and classification accuracy by combining residual learning, attention mechanism, and soft thresholding, making it particularly suitable for image recognition and classification tasks. The ERAN model features an innovative design in feature extraction and data processing. First, by combining residual learning with an attention mechanism, ERAN effectively captures subtle pathological features in MRI images and employs a soft-thresholding technique to optimize feature selection, minimizing noise interference. These enhancements not only improve the model's classification performance but also strengthen its robustness when handling complex medical images. Additionally, ERAN excels in addressing data imbalance, ensuring accurate classification of minority classes and overcoming the prevalent issue of class imbalance in existing methods. Lastly, the model's relatively simple structure conserves computational resources, facilitating its application in practical clinical diagnostics. A limitation of our current approach, however, is that its generalization capabilities on other datasets require further validation.

## 2 Data sources and preprocessing

### 2.1 Dataset description

The experimental data comes from the Alzheimer's disease dataset on the Kaggle website, ethical review and approval are waived for the study, due to the fact that all research data are from open-source datasets. The original data is manually collected from various websites, and each label is validated. The data consists of MRI images which can be used to evaluate Alzheimer's patients at different stages, and four types of images are included: mild dementia, moderate dementia, non-dementia, and very mild dementia, as shown in Fig 1. The data is divided into a training set and a testing set, with 7821 images in the training set and 3550 images in the testing set. The image sizes vary from 4.3KB to 5.3KB. The image size is 176 * 208 pixels, and the distribution of the number of samples in various image datasets is shown in Table 1.

### 2.2 Data preprocessing

This study first adjusted all images to the same size (32*32) and then normalized pixel values to a range of [0,1]. During image preprocessing, corresponding integer labels were assigned to each category of the image for the training process.

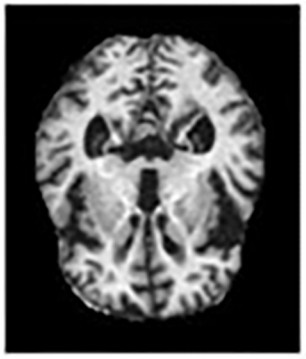

Mild Dementia

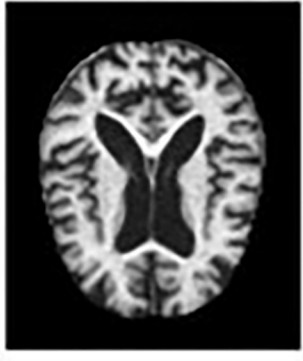

Moderate Dementia

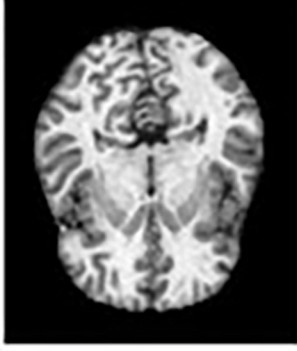

Non-Dementia

Very Mild Dementia

**Fig 1. Alzheimer's disease dataset.**

**Table 1. Distribution of the dataset.**

| Types | Training Set Sample Number | Test Set Sample Number | Total |
|---|---|---|---|
| Mild Dementia | 717 | 179 | 896 |
| Moderate Dementia | 52 | 12 | 64 |
| Non-Dementia | 2560 | 640 | 3200 |
| Very Mild Dementia | 1792 | 448 | 2240 |
| Total | 7821 | 1279 | 9100 |

As there were only 64 samples of moderate dementia, it was insufficient to meet the needs of the deep learning model. In order to increase the number and diversity of training samples, improve the model's generalization ability, and prevent overfitting, this study adopted data augmentation methods to expand each category of image, thereby increasing sample quantity and diversity. Based on the characteristics of Alzheimer's disease image data, this study chose the following augmentation methods: random rotation, random scaling, random translation, and horizontal flipping.

# 3 Enhanced Residual Attention Network (ERAN) model

## 3.1 Model architecture

The detailed explanation of the architecture of the Enhanced Residual Attention Network model is as follows, shown in Fig 2.

1. Initial convolutional layer. This is the first layer of the model, which uses convolution operations to extract primary features of the input image. It is achieved by using a convolutional layer without bias, followed by a batch normalization and ReLU activation function.

2. Residual module. The model consists of multiple residual modules, each performing two convolution operations, followed by a batch normalization and ReLU activation each time. These modules aim to enable the network to learn incremental features and help the network practice more effectively at a deeper level.

3. Attention mechanism module. After each residual module, an SE module is added, which is implemented through global average pooling, reshaping, fully connected layers (first dimensionality reduction and then dimensionality enhancement), and a sigmoid activation function. The purpose of the attention mechanism module is to enhance the network's representation ability by modeling the dependencies between channels. Attention mechanism enables the model to automatically focus on the lesion area, and improves sensitivity to subtle pathological changes.

4. Soft thresholding. Following the attention mechanism module, this architecture uses a custom soft thresholding layer aimed at further purifying feature representation, reducing small amplitude features and enhancing the expression of important features by performing soft thresholding operations on input features. It effectively filters noise, and improves signal-to-noise ratio.

5. Residual connection. At the end of each residual module, the input and processed output are merged through an addition operation, and then activated by a ReLU function. This design helps to alleviate the problem of gradient vanishing and improve the training efficiency of the model at depth.

6. Global average pooling and classification. This module uses a global average pooling layer to reduce dimensionality, and outputs the final classification result through a fully

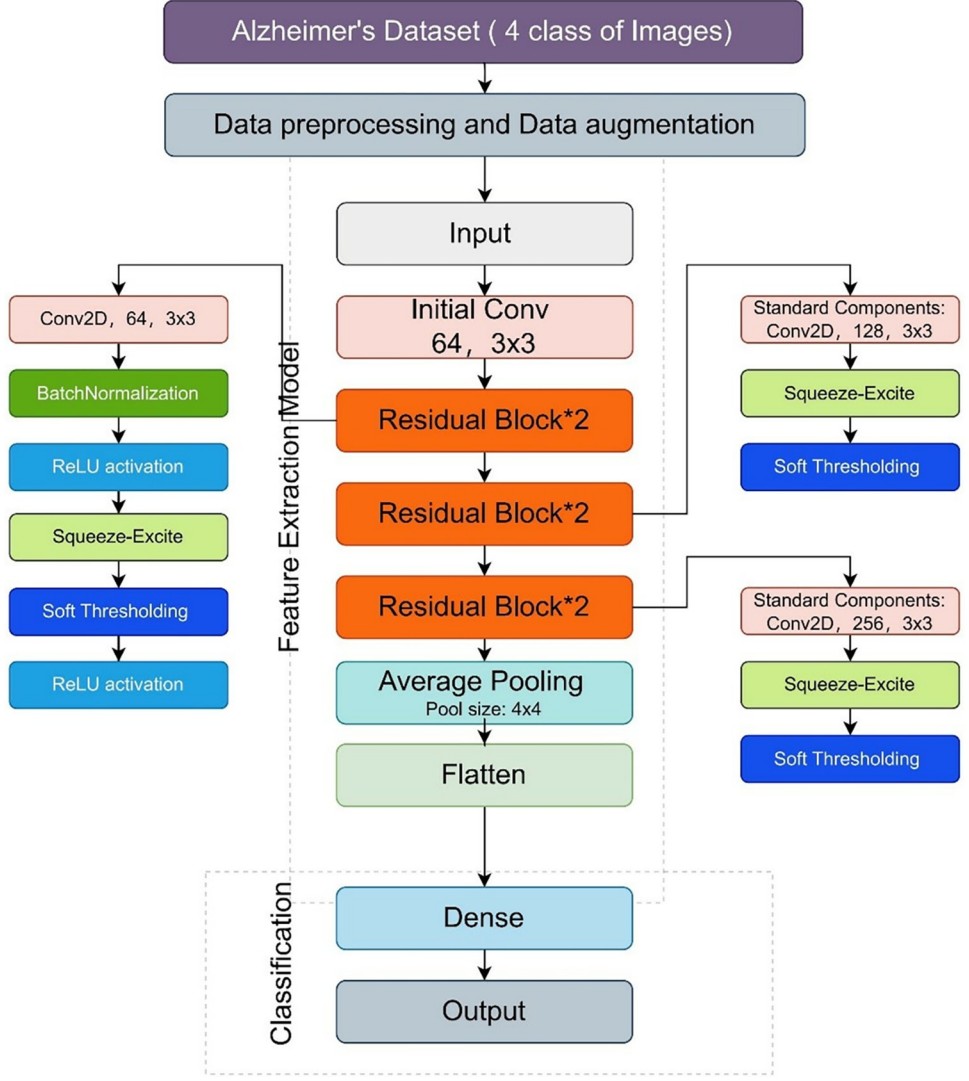

**Fig 2. Enhanced Residual Attention Network (ERAN) model architecture.**

connected layer. This design helps to reduce the number of model parameters while maintaining the global information of features.

7. Output layer. Finally, this module converts the network's output into a probability distribution using a SoftMax activation function for multi class classification tasks.

**3.1.1 Residual module.**   The residual module is a fundamental component of a deep residual network (ResNet), as shown in Fig 3. This module mainly consists of two convolutional layers, which contain batch normalization and nonlinear activation function (ReLU) between them. The key feature of the residual module is to add a skip connection to the convolutional layer, thus adding the input feature map to the convolutional processed feature map. This structure helps to solve the problem of gradient vanishing, allowing the model to undergo deeper training.

The residual module consists of two convolutional layers and an identity mapping. The first convolutional layer uses a 3*3 kernel with a stride of 1 and padding of "same", and after

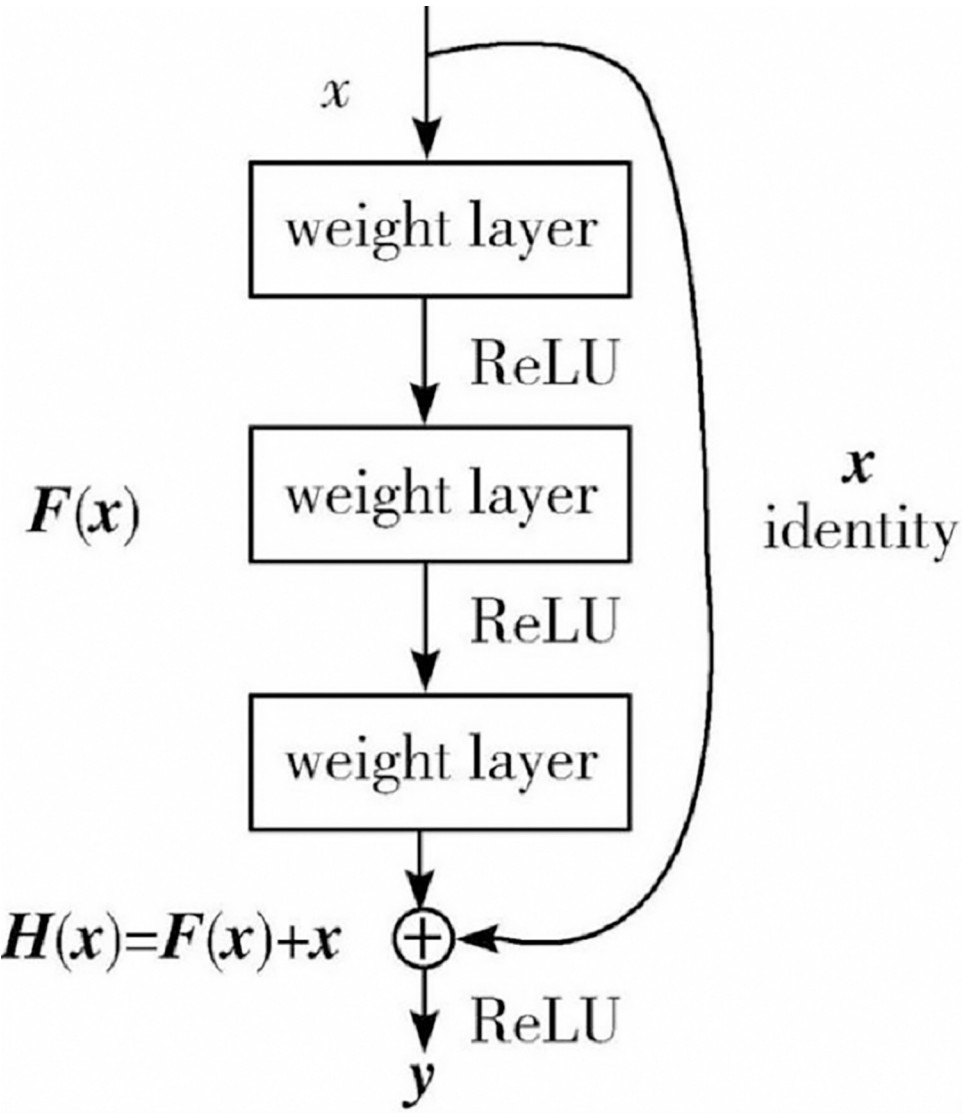

**Fig 3. Standard residual module.**

convolution, batch normalization and ReLU activation function are used. The second convolutional layer also uses a 3*3 kernel with a stride of 1 and padding of "same", and uses batch normalization after convolution. Finally, add the output of the second convolutional layer to the identity map and apply the ReLU activation function.

**3.1.2 Attention module.**    The attention module aims to capture the dependency relationships between different channels, thereby improving the expression ability of features [20]. This study uses a channel attention module to improve model performance. The channel attention module aims to capture channel dependency relationships between input feature maps in order to better focus on important channel information [21, 22].

Firstly, perform Global Average Pooling and Global Max Pooling on the input feature map. Then, pass the results of these two operations separately to a shared Dense Layer. Next, add up the outputs of these two fully connected layers and obtain attention weights through a Sigmoid activation function. Finally, multiply the attention weight with the input feature map to

achieve adaptive weighting of the channel. This allows the model to assign different weights to different channels based on their importance [23–27].

**3.1.3 Soft thresholding.** Soft thresholding is a signal denoising technique that can effectively remove noise and redundant information from features, as shown in Formula (1) [12]. In this experiment, soft thresholding is introduced into the Enhanced Residual Attention Network (ERAN) to achieve adaptive feature selection. The soft thresholding function uses a custom Lambda layer. It takes a threshold parameter and sets certain values in the feature to 0 or shrinks them based on the relationship between the absolute value of the input feature and the threshold. Specifically, if the absolute value of a feature is less than the threshold, the feature value will be set to 0; if the absolute value of the feature is higher than the threshold, the feature value will shrink towards the center. This helps to eliminate noise and redundant data, thereby improving the performance of the model.

$$\mathbf{y} = \begin{cases} x - \tau, & x > \tau \\ 0, & -\tau \leq x \leq \tau \\ x + \tau, & x < -\tau \end{cases} \tag{1}$$

## 3.2 Hyper parameter search and optimization

In order to optimize model performance, this study uses the Hyperband algorithm in the Keras Tuner library to search and optimize hyper parameters. The Keras Tuner library is a library used for hyper parameter search, which can find the optimal combination of hyper parameters in a model. The Hyperband algorithm is an optimization algorithm based on adaptive resource allocation and early termination techniques, with the goal of finding the optimal hyper parameters as much as possible within a limited time. This study uses the Hyperband algorithm to search for the learning rate of the model, which includes 30 iterations. During each iteration, the algorithm selects a value within the given learning rate range (0.00005, 0.01) and trains and evaluates the model. Based on the evaluation results, the algorithm will automatically adjust the search range to make it more likely to find the optimal learning rate in subsequent iterations.

The accuracy and loss comparison of ERAN using different optimizers (Adam, SGD, RMSprop) on the training and validation sets are shown in Fig 4. The graph tells us that the SGD optimizer has the fastest convergence, the best stability, and the best training effect during the training process.

The specific parameters of the experiment are shown in Table 2 in which the maximum number of iterations for the optimal value is 30, the optimizer is SGD, the learning rate of SGD is 0.06297362580630474, shown in Fig 5, and the number of training samples per batch is 32.

## 4 Comparative experiments and results

This part will introduce the experimental process and result analysis, including the experimental environment, comparative experiments, and result analysis. This study will compare the performance of ERAN, Attention_Unet, Tiny-VIT, RSN, and MobileNet V3 models in Alzheimer's disease detection tasks, from two aspects i.e. test accuracy rate and test loss rate.

### 4.1 Experimental environment

This experiment was implemented using the Python programming language and the TensorFlow deep learning framework. The experiment was conducted on the Kaggle website with a

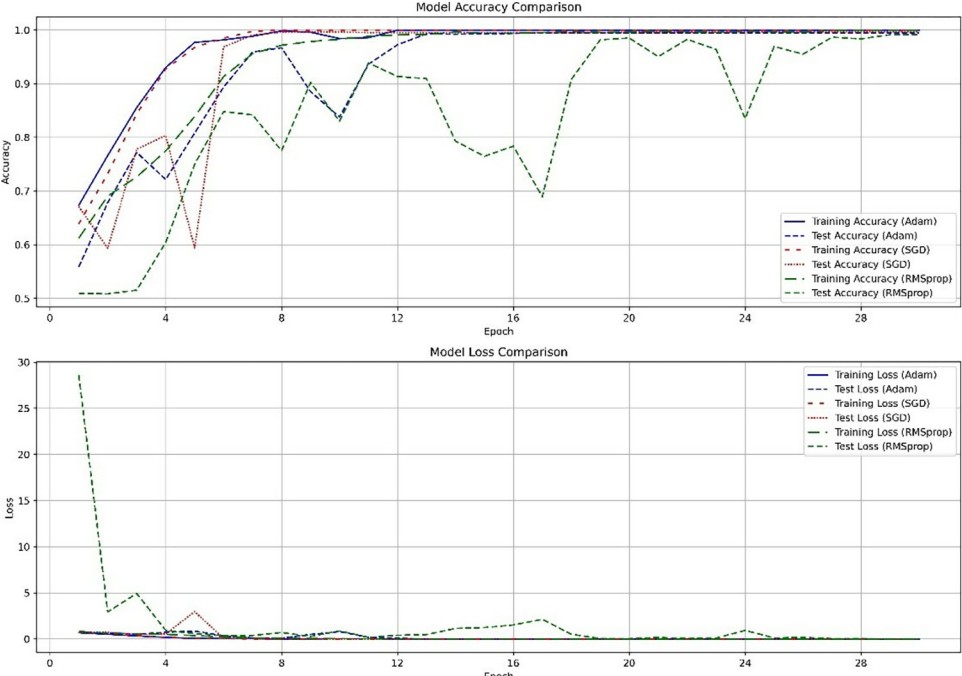

**Fig 4. Comparison of accuracy and loss rates between various optimizer on training and validation sets.**

P100 graphics card. Models such as ERAN, Attention_Unet, Tiny-VIT, RSN, and MobileNet V3 were used for training and validation, and all models adopted the same data processing and training strategy. The training set used in this study utilized the same data augmentation techniques and set the learning rates of all models to the same initial values. Each model was trained for 30 cycles and its performance was evaluated on the test set.

## 4.2 Comparative experiments

To demonstrate the effectiveness of ERAN in Alzheimer's disease detection tasks, comparative experiments are conducted together with four other models, including AttentionUnet, Tiny VIT, RSN, and MobileNet V3. Attention U-Net is a model that combines the U-Net architecture and attention mechanism, aiming to provide more detailed feature learning in the field of image segmentation. By introducing an attention gating mechanism into the standard U-Net structure, this model can better focus on key areas of the image, thereby improving segmentation accuracy. Tiny-VIT, as Tiny Vision Transformer, is a lightweight visual Transformer model suitable for resource limited environments. It adopts the Transformer architecture to process image tasks, captures global dependencies through self-attention mechanism, and it does not rely on traditional convolution operations. Residual Shrinkage Network (RSN)

**Table 2. Hyper parameter table.**

| Network Parameters | Final Determined Value |
|---|---|
| Epochs | 30 |
| Learning-rate | 0.06297362580630474 |
| Optimizer | SGD |
| Batch size | 32 |

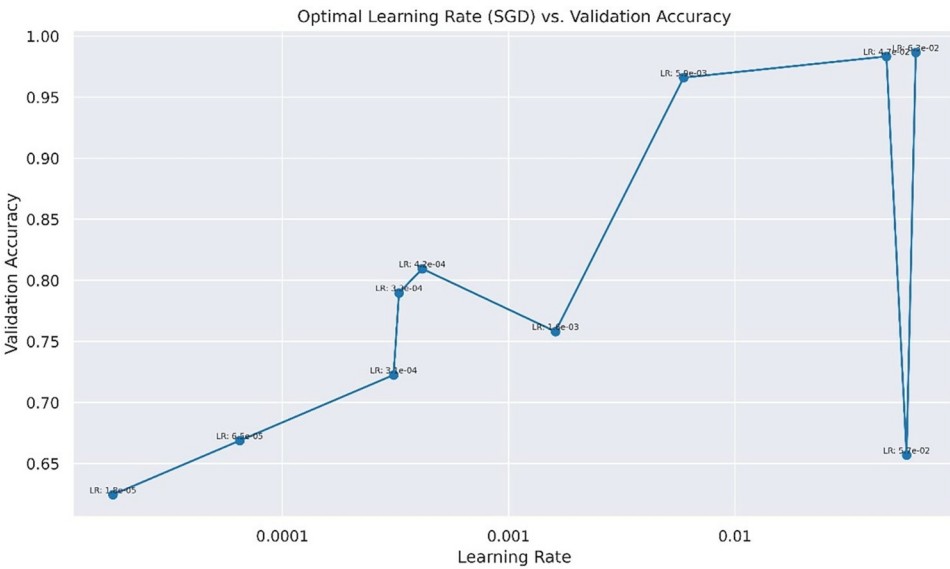

**Fig 5. Comparison of accuracy and learning rate.**

utilizes the concept of residual learning and introduces a novel shrinkage mechanism to reduce model complexity and improve efficiency. By adding a shrinkage step in residual connections, RSN aims to reduce unnecessary features and focus on more informative features, thereby improving the learning efficiency and performance of the model. MobileNet V3 is a light-weight deep learning model optimized for mobile and embedded devices. It reduces computation and model size by using compressed convolutions, SE modules, and H-Swish activation functions, while attempting to maintain high accuracy. During the training and testing process, the Enhanced Residual Attention Network (ERAN) proposed in this study is compared with several other mainstream models, and the accuracy and loss rate of each model on the test set are shown in Table 3.

The best accuracy and loss rate of ERAN on the test set are 0.9936 and 0.0264 respectively. The accuracy and loss rate of ERAN on the training and testing sets are shown in Fig 6.

The best performance of Attention_UNet in terms of accuracy and loss on the test set is 0.9806 and 0.0766 respectively. The accuracy and loss rate of Attention_UNet on the training and testing sets are shown in Fig 7.

The best accuracy and loss rate of Tiny-VIT on the test set are 0.9434 and 0.1584 respectively. The accuracy and loss rate of Tiny-VIT on the training and testing sets are shown in Fig 8.

The Residual Shrinkage Network (RSN) performs the best in terms of accuracy and loss on the test set, with values of 0.9561 and 0.2385 respectively. The accuracy and loss rate of RSN on the training and testing sets are shown in Fig 9.

**Table 3. Accuracy and loss rate of each model.**

| Model | Test Accuracy | Test Loss Rate |
|---|---|---|
| ERAN | 99.36% | 0.0264 |
| Attention_UNet | 98.06% | 0.0766 |
| Tiny-Vit | 94.34% | 0.1584 |
| RSN | 95.61% | 0.2385 |
| MobileNet V3 | 65.35% | 0.8590 |

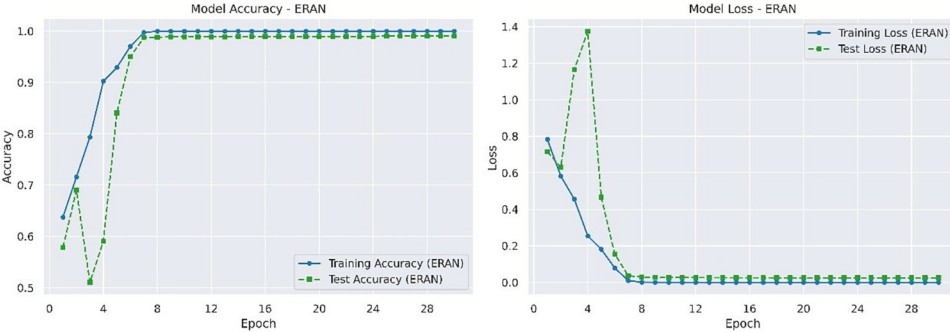

**Fig 6. Accuracy and loss rate of ERAN on training and testing datasets.**

The best accuracy and loss rate of MobileNet V3 on the test set are 0.6355 and 0.8590, respectively. The accuracy and loss rate of MobileNet V3 on the training and testing sets are shown in Fig 10.

## 4.3 Experimental results

We can figure out from Table 3 that ERAN performs the best in this comparative experiment, achieving an accuracy of 99.36% on the test set, owing to its efficient integration of residual learning, attention mechanism, and soft thresholding strategy. Residual learning, a skip connection network, can reduce the number of layers in the model, optimize model redundancy, and make the model lighter. Attention mechanism enables the model to automatically focus on the lesion area, and improves sensitivity to subtle pathological changes. Soft thresholding effectively filters noise, improves signal-to-noise ratio, and ensures that the model focuses more accurately on features. The loss rate of the ERAN model on the test set is significantly lower than other models at 0.0264, indicating that the ERAN model has strong fitting ability for Alzheimer's disease image data and can better capture potential structures and information in the data. The accuracies of Attention U-Net and RSN on the test set reach 98.06% and 95.61% respectively, with loss rates of 0.0766 and 0.2385, indicating its strong ability in focusing on key areas of images and reducing model complexity. Tiny-VIT achieves an accuracy of 94.34% and a loss rate of 0.1584 on the test set, but it provides a feasible solution for resource constrained environments. MobileNet V3 performs the worst on this test dataset, with an accuracy rate of 65.35% and a loss rate of 0.8590, indicating that it cannot effectively extract image features. The experimental results have shown that the ERAN model has high accuracy

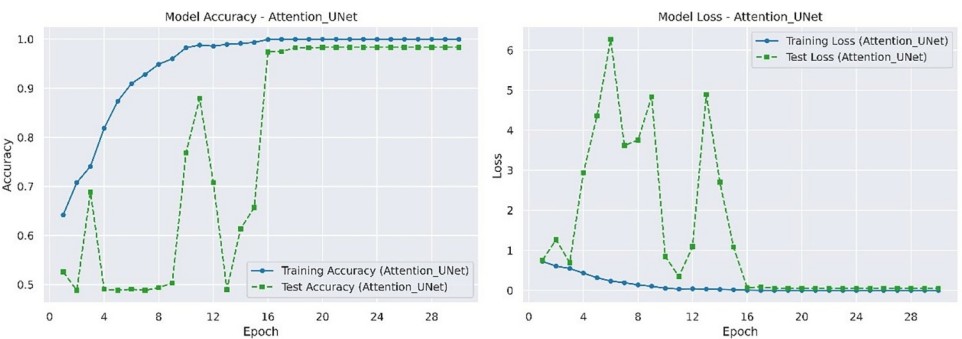

**Fig 7. Accuracy and loss rate of Attention_UNet on training and testing datasets.**

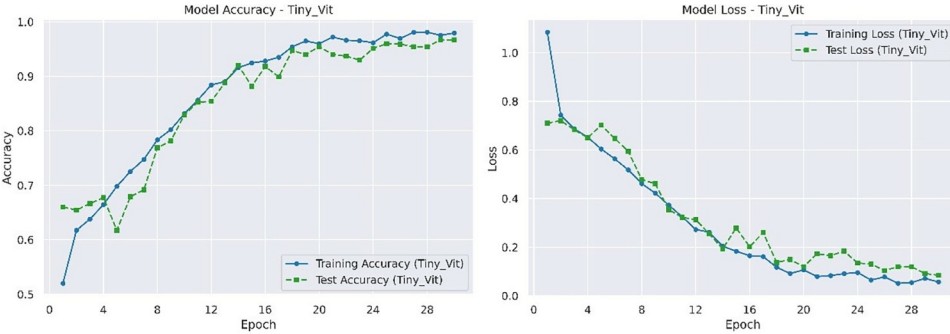

**Fig 8. Accuracy and loss rate of Tiny-VIT on training and testing datasets.**

and low loss rate in Alzheimer's disease detection tasks, making it an efficient and reliable method.

## 5 Conclusion

This study mainly focuses on the detection task of Alzheimer's disease and proposes a method based on Enhanced Residual Attention Network (ERAN). In the experiment, ERAN is compared with models such as Attention_Unet, Tiny VIT, RSN, and MobileNet V3. It's found in the comparative experiments that ERAN exhibits high accuracy and low loss rate in Alzheimer's disease detection tasks, mainly due to its characteristics. ERAN improves the model's feature representation ability and classification accuracy by combining residual learning, attention mechanism, and soft thresholding strategy. The residual module helps alleviate the problem of gradient vanishing, enabling the model to learn more deeply; the attention module enhances the network's representation ability through global average pooling, reshaping, and fully connected layers; and the soft thresholding layer further purifies the feature representation. ERAN has achieved an accuracy of 0.9936 and a loss rate of 0.0264 on the test set, both of which are superior to other models. ERAN's high accuracy and low misdiagnosis rate in AD classification hold significant clinical implications. With an accuracy of 99.36%, ERAN reliably identifies AD patient types, providing a powerful auxiliary tool for clinical diagnosis. Moreover, ERAN's efficiency and low computational demands enhance its operability in real-world clinical settings, enabling rapid and large-scale screening that contributes to overall diagnostic efficiency and coverage. Lastly, ERAN demonstrates the vast potential of deep learning in

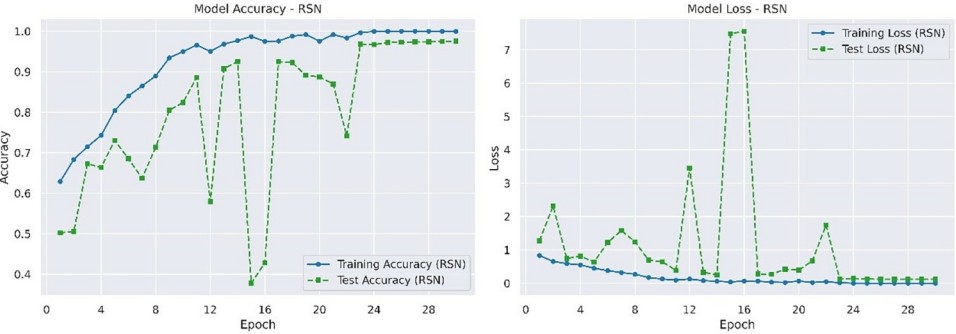

**Fig 9. Accuracy and loss rate of RSN on training and testing datasets.**

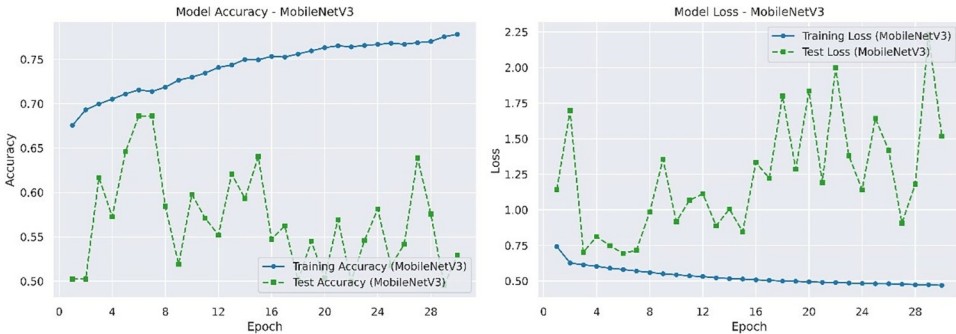

**Fig 10. Accuracy and loss rate of MobileNet V3 on training and testing datasets.**

medical image analysis, advancing the development of intelligent healthcare and offering new approaches and methods for early diagnosis of more complex diseases in the future. The main innovative points and contributions of this study are summarized as follows.

(1) It proposes an Alzheimer's disease detection method based on enhanced residual attention network, which fully utilizes residual modules, attention mechanisms, and soft threshold processing techniques to achieve efficient classification of MRI medical images.

(2) It adopts adaptive algorithms and learning rate adjustment strategies to optimize the model training process and improve the model's generalization ability.

(3) The effectiveness of the proposed method is verified through comparative experiments, and its performance is compared with models such as AttentionUnet, Tiny VIT, RSN, and MobileNet V3, proving that the proposed method has superior performance in Alzheimer's disease detection.

(4) The ERAN model proposed in this study has high accuracy and low loss rate, which has important implications for existing research and plays an important role in future clinical applications, providing strong support for the early diagnosis and treatment of Alzheimer's disease. The method used in this study demonstrates good performance in Alzheimer's disease detection tasks. However, there are also certain limitations, and there is room for improvement in the following areas of this study.

(1) The size of the dataset should be expanded. In order to improve the practicality and performance of the model in practical clinical environments, the model can be trained and validated on larger datasets. This will help evaluate the effectiveness and reliability of the model in practical application scenarios.

(2) Attempt to achieve multimodal data fusion In future research, multiple medical images (such as MRI, PET, etc.) or non-image data (such as genetic data, clinical data, etc.) can be attempted to be fused to improve the accuracy of Alzheimer's disease detection.

## 6 Future work

We wonder if the ERAN proposed in this study has good generalization ability and can be applied to other medical image analysis tasks, such as tumor detection, heart disease diagnosis, etc. In order to provide more accurate and valuable auxiliary diagnostic tools for clinical doctors, further research will be conducted in the future.

## Author Contributions

**Conceptualization:** Xiaoli Li.

**Investigation:** Xiaoli Li, Xinfang Chen, Guoming Yuan.

**Methodology:** Bairui Gong, Xinfang Chen.

**Software:** Bairui Gong.

**Writing – original draft:** Xiaoli Li.

**Writing – review & editing:** Xiaoli Li, Hui Li.

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
