## [Decision Letter · Decision Letter 0]

23 Jun 2024

PONE-D-24-16095Alzheimer's Disease Image Classification Based on Enhanced Residual Attention NetworkPLOS ONE

Dear Dr. Li,

Thank you for submitting your manuscript to PLOS ONE. After careful consideration, we feel that it has merit but does not fully meet PLOS ONE’s publication criteria as it currently stands. Therefore, we invite you to submit a revised version of the manuscript that addresses the points raised during the review process. 

We look forward to receiving your revised manuscript.

Kind regards,

Ian James Martins, PhD

Academic Editor

PLOS ONE

Additional Editor Comments (if provided):

The authors need to ensure that the research is properly verified and need to hone in on the key points raised by the reveiwers and correct inadvertent errors. The revised manuscript must maintain the high standards for peer-reveiwed journals.

Reviewers' comments:

Reviewer's Responses to Questions

**Comments to the Author**

1. Is the manuscript technically sound, and do the data support the conclusions?

Reviewer #1: Partly

Reviewer #2: Yes

2. Has the statistical analysis been performed appropriately and rigorously? 

Reviewer #1: No

Reviewer #2: Yes

3. Have the authors made all data underlying the findings in their manuscript fully available?

Reviewer #1: Yes

Reviewer #2: Yes

4. Is the manuscript presented in an intelligible fashion and written in standard English?

Reviewer #1: No

Reviewer #2: Yes

5. Review Comments to the Author

Reviewer #1: The manuscript has significant potential to contribute positively to the field of Image Analysis for early detection of Alzheimer's disease. However, certain number of revisions are required before the manuscript could be considered for publication.

General comment: The manuscript needs overall correction in terms of language (scientific explanations with a comprehensible flow) and formatting (font types and sizes, equation editing etc).

Specific comments:

1. Detailed critical evaluation of exiting related methods is needed to justify the proposed methodology. The manuscript contains one paragraph that reports some relevant work. However, critical evaluation of the success and drawbacks of recent existing methods is necessary to justify the proposed approach. Some more relevant reference might be missing. Eg: Jiang M, Yan B, Li Y, Zhang J, Li T, Ke W. Image Classification of Alzheimer's Disease Based on External-Attention Mechanism and Fully Convolutional Network. Brain Sci. 2022 Feb 26;12(3):319. doi: 10.3390/brainsci12030319. PMID: 35326275; PMCID: PMC89

2. It is necessary to have a concrete justification for the proposed model and the "enhancement" that makes it possible for the model to work well with the types of images used in the study. An ablation study would make the findings more concrete.

3. It is claimed that "The enhanced residual attention network proposed in this study has good generalization

ability and can be applied to other medical image analysis tasks, such as tumor detection, heart

disease diagnosis, etc., in order to provide more accurate and valuable auxiliary diagnostic tools for

clinical doctors." Is there any strong justification for this claim?

Reviewer #2: The manuscript is technically sound and potentially sound for the scientific community. However, methodological details should be included, and the discussion must be improved before acceptance.

1. Provide a table of existing literature methods with research gaps for clearly mapping.

2. The research should address the interpretability of the models' decision-making process.

3. The introduction effectively sets the stage for your research, clearly stating the problem and situating it within the broader field of deep learning and computer vision. It could be enhanced by briefly discussing previous key studies to provide a richer background.

4. Some papers can be discussed. For example:

- https://doi.org/10.1007/s11571-024-10104-1

- https://doi.org/10.1007/s11571-022-09859-2

- https://doi.org/10.1016/j.compbiomed.2021.104828

5. The conclusion successfully encapsulates the study's contributions but could be enhanced by discussing potential future research directions or applications of your findings in a real-world context.

6. You should give a proper discussion section and discussions should be increased. Limitations and benefits should be given in bullets.

6. PLOS authors have the option to publish the peer review history of their article (what does this mean?). If published, this will include your full peer review and any attached files.

Reviewer #1: No

Reviewer #2: No

---

## [Author Response · Author response to Decision Letter 0]

12 Jul 2024

Reviewer #1: The manuscript has significant potential to contribute positively to the field of Image Analysis for early detection of Alzheimer's disease. However, certain number of revisions are required before the manuscript could be considered for publication.

General comment: The manuscript needs overall correction in terms of language (scientific explanations with a comprehensible flow) and formatting (font types and sizes, equation editing etc).

The language and format have been modified.

Specific comments:

1. Detailed critical evaluation of exiting related methods is needed to justify the proposed methodology. The manuscript contains one paragraph that reports some relevant work. However, critical evaluation of the success and drawbacks of recent existing methods is necessary to justify the proposed approach. Some more relevant reference might be missing. Eg: Jiang M, Yan B, Li Y, Zhang J, Li T, Ke W. Image Classification of Alzheimer's Disease Based on External-Attention Mechanism and Fully Convolutional Network. Brain Sci. 2022 Feb 26;12(3):319. doi: 10.3390/brainsci12030319. PMID: 35326275; PMCID: PMC89

A critical evaluation of the success and shortcomings of existing methods has been conducted in the article. Thank you for providing this paper on Alzheimer's image classification based on deep learning. This is indeed an important research achievement that has been carefully read and cited in this paper.

2. It is necessary to have a concrete justification for the proposed model and the "enhancement" that makes it possible for the model to work well with the types of images used in the study. An ablation study would make the findings more concrete.

Thank you for providing the research method. The ERAN model is based on ResNet, combining attention mechanism and soft thresholding, as explained in the paper. Is this considered a form of ablation research? Due to time constraints, more detailed ablation study of this model will be used in future research.

3. It is claimed that "The enhanced residual attention network proposed in this study has good generalization ability and can be applied to other medical image analysis tasks, such as tumor detection, heart disease diagnosis, etc., in order to provide more accurate and valuable auxiliary diagnostic tools for clinical doctors." Is there any strong justification for this claim?

In order to verify the good generalization ability of the model proposed in this study, we classified tumor images using the model and conducted comparative experiments. As shown in FIG 1 in the doc response to reviewer1. The experimental results showed that the model had better performance than other models and had good generalization ability. At the same time, this statement was included in the future research section for further verification in future studies.

Reviewer #2: The manuscript is technically sound and potentially sound for the scientific community. However, methodological details should be included, and the discussion must be improved before acceptance.

1. Provide a table of existing literature methods with research gaps for clearly mapping.

Sengul Dogan's research paper on EEG signals fills the research gap, and the relevant paper has been cited in this article.

2. The research should address the interpretability of the models' decision-making process.

This paper combines residual learning, attention mechanism, and soft thresholding, to improve the accuracy of image classification.

3. The introduction effectively sets the stage for your research, clearly stating the problem and situating it within the broader field of deep learning and computer vision. It could be enhanced by briefly discussing previous key studies to provide a richer background.

Some key researches have been added in this paper.

4. Some papers can be discussed. For example:

- https://doi.org/10.1007/s11571-024-10104-1

- https://doi.org/10.1007/s11571-022-09859-2

- https://doi.org/10.1016/j.compbiomed.2021.104828

The above papers have been discussed and cited in the study. 

5. The conclusion successfully encapsulates the study's contributions but could be enhanced by discussing potential future research directions or applications of your findings in a real-world context.

“Future work” has been added in this paper.

6. You should give a proper discussion section and discussions should be increased. Limitations and benefits should be given in bullets.

We have developed the discussion section and explored potential future research directions. Limitations and benefits have been given in the paper.

---

## [Decision Letter · Decision Letter 1]

5 Oct 2024

PONE-D-24-16095R1Alzheimer's disease image classification based on enhanced residual attention networkPLOS ONE

Dear Dr. Li,

Thank you for submitting your manuscript to PLOS ONE. After careful consideration, we feel that it has merit but does not fully meet PLOS ONE’s publication criteria as it currently stands. Therefore, we invite you to submit a revised version of the manuscript that addresses the points raised during the review process.

We look forward to receiving your revised manuscript.

Kind regards,

Ian James Martins

Academic Editor

PLOS ONE

Journal Requirements:

Comments from PLOS Editorial Office: We note that one or more reviewers has recommended that you cite specific previously published works. As always, we recommend that you please review and evaluate the requested works to determine whether they are relevant and should be cited. It is not a requirement to cite these works. We appreciate your attention to this request.

Reviewers' comments:

Reviewer's Responses to Questions

**Comments to the Author**

1. If the authors have adequately addressed your comments raised in a previous round of review and you feel that this manuscript is now acceptable for publication, you may indicate that here to bypass the “Comments to the Author” section, enter your conflict of interest statement in the “Confidential to Editor” section, and submit your "Accept" recommendation.

Reviewer #1: (No Response)

Reviewer #2: All comments have been addressed

Reviewer #3: All comments have been addressed

Reviewer #4: All comments have been addressed

2. Is the manuscript technically sound, and do the data support the conclusions?

Reviewer #1: Partly

Reviewer #2: Yes

Reviewer #3: Yes

Reviewer #4: Yes

3. Has the statistical analysis been performed appropriately and rigorously? 

Reviewer #1: I Don't Know

Reviewer #2: Yes

Reviewer #3: Yes

Reviewer #4: Yes

4. Have the authors made all data underlying the findings in their manuscript fully available?

Reviewer #1: No

Reviewer #2: Yes

Reviewer #3: Yes

Reviewer #4: Yes

5. Is the manuscript presented in an intelligible fashion and written in standard English?

Reviewer #1: Yes

Reviewer #2: Yes

Reviewer #3: (No Response)

Reviewer #4: Yes

6. Review Comments to the Author

Reviewer #1: (No Response)

Reviewer #2: I have appreciated the deep revision of the contents and the present form of this manuscript. All my previous concerns have been accurately addressed. I think that this paper can be accepted.

Reviewer #3: The issues are listed in the following:

1. The professional English editing is recommended. The authors should get editing help from someone with full professional proficiency in English.

2. The abstract should highlight the key objectives, methodology, and outcomes clearly. Cite some high quality or relatively new literature, For example, CNN's method introduction can cite "A Novel Centralized Federated Deep Fuzzy Neural Network with Multi-objectives Neural Architecture Search. For Epistatic Detection, DOI: 10.1109 / tfuzz. 2024.3369944 ", The introduction of image segmentation methods can be quoted in "Soft Attention Based DenseNet Model for Parkinson's Disease Classification Using SPECT Images, DOI10.3389 / fnagi. 2022.908143"

3. It is recommended to add an in-depth analysis of the successes and limitations of existing methods and discuss how these compare with your proposed method.

4. The manuscript needs a concrete justification for the proposed model, particularly the “enhancement” that enables it to work well with the types of images used in the study.

5. It is suggested to provide more experimental results or case studies to demonstrate the performance of ERAN on different types of medical image data.

6. Provide more charts and visualization tools to clearly show the performance comparison of ERAN and other models on the test set.

7. Conduct an in-depth analysis of the experimental results and explain the reasons for the superior performance of ERAN.

8. Strengthen the discussion section by providing an in-depth interpretation of the results and their significance to existing research and clinical practice.

9. Discuss the limitations of the study and propose future research directions.

10. High-resolution images should be provided to ensure that all details are visible and can be thoroughly examined by readers.

Reviewer #4: This study introduces a deep learning model, the Enhanced Residual Attention Network (ERAN), designed to detect Alzheimer's disease by classifying medical images. By integrating residual learning, attention mechanisms, and soft thresholding, the model's ability to represent features and its classification accuracy have been enhanced. The model achieved an accuracy of 99.36% in detecting Alzheimer's disease, with a loss rate of just 0.0264. These experimental results demonstrate that ERAN performs exceptionally well on the Alzheimer's disease test dataset, offering valuable support for early diagnosis and treatment.

The authors have made all the corrections suggested by the reviewers.

7. PLOS authors have the option to publish the peer review history of their article (what does this mean?). If published, this will include your full peer review and any attached files.

Reviewer #1: No

Reviewer #2: No

Reviewer #3: No

Reviewer #4: No

---

## [Author Response · Author response to Decision Letter 1]

9 Oct 2024

Thank you for your review of the paper. We have made revisions to the paper according to your guidance and suggestions. Please check.

---

## [Decision Letter · Decision Letter 2]

7 Nov 2024

PONE-D-24-16095R2Alzheimer's disease image classification based on enhanced residual attention networkPLOS ONE

Dear Dr. Li,

Thank you for submitting your manuscript to PLOS ONE. After careful consideration, we feel that it has merit but does not fully meet PLOS ONE’s publication criteria as it currently stands. Therefore, we invite you to submit a revised version of the manuscript that addresses the points raised during the review process. **Please ensure that the research is properly verified and hone in the key points raised by the reviewers and correct inadvertent errors** Please submit your revised manuscript by Dec 22 2024 11:59PM. If you will need more time than this to complete your revisions, please reply to this message or contact the journal office at plosone@plos.org. Please include the following items when submitting your revised manuscript:A rebuttal letter that responds to each point raised by the academic editor and reviewer(s). You should upload this letter as a separate file labeled 'Response to Reviewers'.A marked-up copy of your manuscript that highlights changes made to the original version. You should upload this as a separate file labeled 'Revised Manuscript with Track Changes'.An unmarked version of your revised paper without tracked changes. You should upload this as a separate file labeled 'Manuscript'.

We look forward to receiving your revised manuscript.

Kind regards,

Ian James Martins, PhD

Academic Editor

PLOS ONE

Reviewers' comments:

Reviewer's Responses to Questions

**Comments to the Author**

1. If the authors have adequately addressed your comments raised in a previous round of review and you feel that this manuscript is now acceptable for publication, you may indicate that here to bypass the “Comments to the Author” section, enter your conflict of interest statement in the “Confidential to Editor” section, and submit your "Accept" recommendation.

Reviewer #1: (No Response)

Reviewer #3: All comments have been addressed

2. Is the manuscript technically sound, and do the data support the conclusions?

Reviewer #1: No

Reviewer #3: Yes

3. Has the statistical analysis been performed appropriately and rigorously? 

Reviewer #1: No

Reviewer #3: Yes

4. Have the authors made all data underlying the findings in their manuscript fully available?

Reviewer #1: No

Reviewer #3: Yes

5. Is the manuscript presented in an intelligible fashion and written in standard English?

Reviewer #1: Yes

Reviewer #3: Yes

6. Review Comments to the Author

**Reviewer #1: **The reviewer comments have not been sufficiently addressed in the revisions made by the authors. For example;

1. A detailed analysis of the successes and limitations of existing methods and how they compare with the proposed method

2. Providing an in-depth interpretation of the results and their significance to existing research and clinical practice

Above are essential to justify the validity of the proposed work.

Therefore it is recommended to resubmit after a full revision addressing the key issues.

**Reviewer #3: **All my problems have been solved, and I propose employment. However, there are still some problems with the format, which should be carefully modified before publication.

7. PLOS authors have the option to publish the peer review history of their article (what does this mean?). If published, this will include your full peer review and any attached files.

Reviewer #1: No

Reviewer #3: No

---

## [Author Response · Author response to Decision Letter 2]

16 Nov 2024

Response to reviewer #1: 

1. A detailed analysis of the successes and limitations of existing methods and how they compare with the proposed method

A detailed analysis of the strengths and limitations of existing methods, along with a comparison to the proposed ERAN model, is provided in the introduction section.

2. Providing an in-depth interpretation of the results and their significance to existing research and clinical practice

An in-depth interpretation of the results and their implications for existing research and clinical practice is discussed in the conclusion section.

---

## [Decision Letter · Decision Letter 3]

27 Dec 2024

Alzheimer's disease image classification based on enhanced residual attention network

PONE-D-24-16095R3

Dear Dr.Xiaoli Li,

We’re pleased to inform you that your manuscript has been judged scientifically suitable for publication and will be formally accepted for publication once it meets all outstanding technical requirements.

The authors have ensured that the research is properly verified to maintain the high standards for peer-reveiwed journals .

Kind regards,

Ian James Martins, PhD

Academic Editor

PLOS ONE

Additional Editor Comments (optional):

Reviewers' comments:

Reviewer's Responses to Questions

**Comments to the Author**

1. If the authors have adequately addressed your comments raised in a previous round of review and you feel that this manuscript is now acceptable for publication, you may indicate that here to bypass the “Comments to the Author” section, enter your conflict of interest statement in the “Confidential to Editor” section, and submit your "Accept" recommendation.

Reviewer #2: All comments have been addressed

Reviewer #3: All comments have been addressed

2. Is the manuscript technically sound, and do the data support the conclusions?

Reviewer #2: Yes

Reviewer #3: Yes

3. Has the statistical analysis been performed appropriately and rigorously? 

Reviewer #2: (No Response)

Reviewer #3: Yes

4. Have the authors made all data underlying the findings in their manuscript fully available?

Reviewer #2: Yes

Reviewer #3: Yes

5. Is the manuscript presented in an intelligible fashion and written in standard English?

Reviewer #2: Yes

Reviewer #3: Yes

6. Review Comments to the Author

Reviewer #2: The authors took a professional approach to the suggestions sent in advance and put in a lot of work to refine the content of the article. I believe that the article has benefited from this scientifically and the content is clearer for the reader. The article in its current form is interesting, well structured and the subject matter is of topical importance, and I therefore recommend its publication.

Reviewer #3: This edition has solved all my doubts and has been revised accordingly, and is proposed for publication in this journal.

7. PLOS authors have the option to publish the peer review history of their article (what does this mean?). If published, this will include your full peer review and any attached files.

Reviewer #2: No

Reviewer #3: No

---

## [Editor Report · Acceptance letter]

16 Jan 2025

PONE-D-24-16095R3 

PLOS ONE

Dear Dr. Li, 

I'm pleased to inform you that your manuscript has been deemed suitable for publication in PLOS ONE. Congratulations! Your manuscript is now being handed over to our production team.

Kind regards, 

on behalf of

Dr. Ian James Martins 

Academic Editor

PLOS ONE